# The Effect of Multistage Refinement on the Bio-Physico-Chemical Properties and Gel-Forming Ability of Fish Protein Isolates from Mackerel (*Rastrelliger kanagurta*)

**DOI:** 10.3390/foods12213894

**Published:** 2023-10-24

**Authors:** Panumas Somjid, Manat Chaijan, Saroat Rawdkuen, Lutz Grossmann, Worawan Panpipat

**Affiliations:** 1Food Technology and Innovation Research Center of Excellence, School of Agricultural Technology and Food Industry, Walailak University, Nakhon Si Thammarat 80160, Thailand; panumas.s14@gmail.com (P.S.); cmanat@wu.ac.th (M.C.); 2Food Science and Technology Program, School of Agro-Industry, Mae Fah Luang University, Chiang Rai 57100, Thailand; saroat@mfu.ac.th; 3Department of Food Science, University of Massachusetts Amherst, 102 Holdsworth Way, Amherst, MA 01002, USA; lkgrossmann@umass.edu

**Keywords:** myofibrillar proteins, pH-shift process, protein isolate, gel

## Abstract

The objective of this research was to improve the protein extraction processes of *Rastrelliger kanagurta* (Indian mackerel) to generate protein isolate with enhanced bio-physico-chemical properties and gel-forming ability. To achieve this, two novel approaches were designed that utilized an additional alkaline separation step and were compared to a conventional process: acid solubilization → alkaline solubilization → pI and acid solubilization → pI → alkaline solubilization. The novel extraction designs resulted in a lower lipid content, lipid oxidation, and TCA-soluble peptides, as well as improving the color and sensory features of the refined proteins, which corresponded to the lowest total heme pigments (*p* < 0.05). Furthermore, the protein isolate recovered with the modified processes showed significant changes in biochemical properties (decreases in Ca^2+^-ATPase activity/reactive sulfhydryl content and an increase in surface hydrophobicity) and dynamic rheological behavior. As a result, by altering the extraction procedure it was possible to obtain improved gel characteristics such as gel strength, color, expelled moisture, and improved gel microstructure. Moreover, this study demonstrated that the gel network was partly stabilized by disulfide bonds, according to SDS-PAGE. Overall, this study demonstrates that by optimizing protein extraction procedures a considerable improvement in quality can be achieved and that an additional alkaline extraction after isoelectric point precipitation results in the optimized gel-forming ability of mackerel proteins.

## 1. Introduction

Fish and fish products are always excellent sources of protein and nutrients for human consumption, with a protein digestibility-corrected amino acid score (PDCAAS) close to 1 [1]. The high nutritional quality and ubiquitous presence of fish around the world have led to the development of fisheries all over the world, including wild fishing and aquaculture [2]. According to the FAO [3], fishery production totaled 179 million tons in 2020, of which 157 million tons were used for human consumption. This overall high demand for fish and fish products has put a lot of strain on many global fish stocks due to overfishing in many regions [4]. However, Indian mackerel (*Rastrelliger kanagurta*) is an underexploited dark-fleshed saltwater fish species that is widely distributed and proliferated in Southern Thailand, which has remained abundant with stable populations over the last decade [5]. This provides us with an opportunity to use Indian mackerel more often as a food product or as a food ingredient to fulfill the market’s primary demand for fish resources while preventing overfishing [6,7]. However, the use of this type of fish has proven to be very difficult, especially in the production of restructured seafood. The main reason for this is that Indian mackerel consists mainly of dark muscle (just beneath the skin along the lateral line) [7]. Because dark muscle contains more pigments and lipids, it is extremely challenging to separate these unwanted components from the protein fraction, which then can be applied as a functional ingredient to produce restructured seafood. As a result, the majority of gel-based products derived from this fish exhibit poor gelling properties and exhibit considerable discoloration and off-flavors derived from lipid oxidation [8].

To overcome these issues, protein extraction using the pH-shift procedure has been proposed as a potential method to increase the functional properties and decrease the presence of off-flavors in this fish [2,9,10]. In brief, the pH-shift method consists of three main steps: (i) protein solubilization at a high (≥10.5 for an alkaline process) or low (≤3.5 for an acidic process) pH, at which the protein carries the highest net charge; (ii) the removal of the water-insoluble fraction consisting of fat, insoluble proteins, and other impurities using a high-speed/decanter centrifuge; and (iii) the precipitation of the soluble protein fraction at the isoelectric point (pH~5.5) [11,12]. Previous researchers have found that this method results in a high protein yield with protein contents in the final powder of ≥90%, thus qualifying it as a protein isolate [4,13,14]. Additionally, the obtained protein isolates had low lipid (<1%) and heme pigment levels [15,16]. Furthermore, both acid and alkali pH-shift processes were reported to have equal and sometimes significantly better gelation characteristics than the typical surimi processing technique. The recovered protein can be used as the main ingredient in many food formulations and functional fish protein gels [17], such as fish sausage [18], fish balls [19], and fish burgers [20], to replace surimi or minced fish.

However, when comparing the acid and alkaline pH-shift processes, each method has its own advantages and limitations. First, acid-solubilization generally results in a higher yield. Second, alkaline-solubilization often leads to less lipid oxidation of the protein isolate [21]. Higher protein yields and lower pigment levels have been more often reported as advantages of the acid-solubilization method, which has been used for several species of fish. An alkaline solubilization either in-process or in the final isolate, can aid the heme/lipid removal efficiency and thereby enhance the gel quality of the resulting protein isolate [22,23,24]. Although the pH-shift treatment effectively reduced lipids and pro-oxidative heme proteins in dark-fleshed fish material to low levels, the remaining residues were still sufficient to induce lipid oxidation, resulting in adverse effects on the color and sensory attributes of the recovered protein [25]. Some studies observed that lipid and heme protein removal was less efficient and the lipid oxidation rate of the recovered protein extract was higher when the acid process was used compared to the alkaline process [26,27]. In addition, most scientists agree that an alkaline-made protein isolate provides superior gel strength compared to an isolate that is produced using the acid-solubilization route [4,10].

From the previous discussion, it can be concluded that both extraction routes have specific advantages when obtaining a fish protein isolate. However, they are not usually combined to leverage the advantages of both approaches. Accordingly, novel processing approaches to improve the efficacy of the acid pH-shift, as well as the prevention of lipid oxidation during processing, are required to improve the quality of the produced fish protein isolate. As a consequence, the current study aims to investigate the effect of a modified acid pH-shift procedure involving post-alkaline processing, simply called the “hybrid pH-shift process”, as a refinement strategy to improve the bio-physico-chemical properties and gel-forming ability of a protein isolate from Indian mackerel.

## 2. Materials and Methods

### 2.1. Fish Sample Preparation

Indian mackerel (*Rastrelliger kanagurta*) with a weight of 65–70 g were bought from a local market in Thasala, Nakhon Si Thammarat, Southern Thailand, and transported to the laboratory in a sealed foamed polystyrene box with a fish/ice ratio of 1:2 (*w*/*w*) within 30 min. The fish were then manually headed, eviscerated, washed, filleted, and skinned, in turn. Then, the fish meat was uniformly minced in a Panasonic MK-G20NR-W grinder (Osaka, Japan). All the steps were conducted on ice or in a cold room at 4 °C, which ensured that the product temperature remained below 5 °C.

### 2.2. Protein Isolation Using Conventional and Modified Acid pH-Shift Processes

The mackerel mince was used to produce protein isolates using three different approaches, following the flow chart in Figure 1. The modified acid pH-shift (acid → pI) processes were divided into two versions with different processes, namely acid → alkaline → pI (AAP) and acid → pI → alkaline (APA) refinement.

Conventional acid pH-shift process: The method for producing the protein isolate was implemented as described by Marmon and Undeland [28]. In brief, mackerel was homogenized with cold distilled water at a 1:9 (*w*/*v*) ratio for 1 min at 20,000 rpm using an IKA^®^ homogenizer (Model T25 digital UltraTurrax^®^, Staufen, Germany). The pH of the homogenate was adjusted to 2.5 using 2 M HCl (constant stirring/10 min). The slurry was then centrifuged (8000× *g*/20 min/4 °C) using a Sorvall Legend XTR Centrifuge (Thermo Fisher Scientific Inc., Yokohama, Japan). The supernatant I was separated from the insoluble fraction after the 1st centrifugation step, and the supernatant was filtered through three layers of cotton sheets. Subsequently, the filtered supernatant I was collected by adjusting the pH to 5.5 using 2 M NaOH. Subsequently, a 2nd centrifugation step was carried out to recover the precipitated proteins, yielding the protein isolate.

Acid → alkaline → pI (AAP) process: After obtaining the supernatant I at pH 2.5 from the conventional acid pH-shift process, the pH of the supernatant I was adjusted to 12.5 using 2 M NaOH, followed by the 2nd centrifugation. After that, the pH of supernatant II was adjusted to 5.5 using 2 M HCl, and the 3rd centrifugation was applied to collect the protein isolate.

Acid → pI → alkaline (APA) process: This process was identical to the conventional acid pH-shift process, except that once the protein isolate was obtained, the alkaline pH-shift process (solubilization at pH 12.5 and precipitation at pH 5.5) was used to further remove residual impurities.

The pH of all recovered proteins was brought to pH 7.0 using 2 N NaOH and all experiments were carried out at this pH. Lastly, the moisture content of the untreated mince (fresh minced fish) and protein isolates was standardized to an average of 80% before being placed in a zip-lock bag and kept in the blast freezer at −18 °C for further analysis (within 1 month).

### 2.3. Characterization of Protein Isolates and Untreated Mince

#### 2.3.1. Determination of Ca^2+^-ATPase Activity, Reactive Sulfhydryl (SH) Group, and Protein Surface Hydrophobicity

The Ca^2+^-ATPase activity of natural actomyosin (NAM) from protein isolates and untreated mince was evaluated using Benjakul et al.’s method [29] and was reported as µmol inorganic phosphate released/mg protein/min.

Ellman’s reagent (5, 50-dithiobis (2-nitrobenzoic acid); DNTB) was used to determine reactive SH content according to Ellman’s method [30]. The reactive SH content was estimated from the absorbance using the molar extinction of 13,600 M^−1^cm^−1^ and was reported as mol/10^8^ g protein.

Surface hydrophobicity of protein was analyzed using the bromophenol blue (BPB) binding method [31] and was reported as the amount of BPB bound (µg).

#### 2.3.2. Determination of TCA-Soluble Peptide

The amount of TCA-soluble peptide, used as an indication of proteolysis, was reported as μmol tyrosine/g sample using the method described by Panpipat and Chaijan [32].

#### 2.3.3. Determination of Total Lipid Content and Thiobarbituric Acid Reactive Substances (TBARS)

The lipid content in untreated mince and protein isolates was evaluated using Bligh and Dyer’s method [33] and was reported as g/100 g sample. The percentage of lipid reduction was calculated relative to the initial lipid content of untreated mince.

The TBARS assay was performed as described by Panpipat et al. [34] and was reported as mg malondialdehyde (MDA) equivalent/kg sample.

#### 2.3.4. Determination of Fishy Odor

The fishy odor of untreated mince and protein isolates was investigated using an experimental methodology endorsed by Walailak University’s Human Research Ethics Committee (WUEC-21-125-01). Untreated mince and protein isolates were formed into patties with a diameter of around 7 cm and placed in a sealed white plastic cup. All samples were conditioned at room temperature for 30 min before being sniffed by 10 trained panelists. The intensity of the fishy odor was scored using a linear scale ranging from none (score = 0) to strong (score = 10).

#### 2.3.5. Determination of Total Heme Protein Content, Myoglobin Absorption Spectra, and Color Characteristics

For total heme protein content, Chaijan and Undeland’s method [35] was used and was reported as µmol hemoglobin/kg of sample.

Myoglobin absorption spectra in the Soret region (350–450 nm) were determined from the supernatant after the precipitation of protein isolates [36].

Color characteristics of the untreated mince, protein isolates, and gels were determined following the method adopted by Panpipat and Chaijan [23]. The values of lightness (*L**), redness/greenness (*a**), and yellowness/blueness (*b**) were determined and the whiteness value was calculated using the following equation [36]:Whiteness = 100 − [(100 − *L**)^2^ + *a**^2^ + *b**^2^]^1/2^
(1)

#### 2.3.6. Sodium Dodecyl Sulfate-Polyacrylamide Gel Electrophoresis (SDS-PAGE)

Protein patterns of all samples were studied via SDS-PAGE, using Laemmli’s method [37].

### 2.4. Gelation

#### 2.4.1. Dynamic Rheological Analysis

The rheological behavior of the untreated mince and protein isolates was measured using a rheometer (HAAKE MARS 60; Thermo Fisher Scientific Inc., Yokohama, Japan) with a 35 mm diameter parallel plate [36]. The temperature sweep was performed from 20 °C to 90 °C with a rate of 2 °C/min to induce gelation, with a constant frequency of 1 Hz and an amplitude strain of 2%. The elastic modulus (G′), viscous modulus (G″), and loss tangent (tan δ) were reported.

#### 2.4.2. Gel Preparation

After being defrosted in a 4 °C refrigerator for 12 h, samples were mixed with 2.5% NaCl (*w*/*w*) using a mortar and pestle for 10 min to form a paste. Subsequently, the pastes were stuffed into a cellulose casing (diameter of 2.5 cm) and tightly sealed. The samples were then set in a water bath at 40 °C for 30 min, followed by cooking at 90 °C for 20 min, then transferred to an ice bath for 30 min, and stored overnight at 4 °C before analysis.

### 2.5. Gel Characterization of Protein Isolates and Untreated Mince

#### 2.5.1. Breaking Force, Deformation, Gel Strength, and Texture Profile Analysis (TPA)

The textural properties of the gel sample were evaluated using a texture analyzer (LR 5K; LLOYD Instruments, West Sussex, UK) via a puncture and TPA. The breaking force (g) and deformation (mm) were measured. Then, the gel strength (g.mm) was calculated [6].

The TPA of gel samples was subjected to a two-cycle compression. Hardness, springiness, cohesiveness, and gumminess were recorded [6].

#### 2.5.2. Expelled Moisture

The water-holding capacity under stress, also known as the expelled moisture content, was determined according to Chaijan et al.’s method [8].

#### 2.5.3. Fourier Transform Infrared (FTIR) Spectra

The gel samples were freeze-dried using a freeze-dryer (FTS systems Inc., Stone Ridge, NY, USA). The FTIR spectra of freeze-dried gel samples were then obtained using a Bruker INVENIO-S FTIR spectrometer (Bruker Co., Ettlingen, Germany) as described by Somjid et al. [6].

#### 2.5.4. Scanning Electron Microscopic (SEM) Images

The microstructure of gels was observed using SEM (GeminiSEM; Carl Ziess Microscopy, Germany) as described by Somjid et al. [38], with an acceleration voltage of 2 kV and with 10,000× magnification.

### 2.6. Statistical Analysis

The data were processed and plotted using GraphPad Prism 8.0 (GraphPad Software Inc., Boston, MA, USA, 2018) and were represented as the means ± SD. All data were analyzed through one-way ANOVA using SPSS 16.0 for Windows (SPSS Inc., Chicago, IL, USA). Comparison of means was performed using Duncan’s multiple-range test. Except for fishy odor analysis and TPA, which were scored by 10 panelists and 6 determinations, all analyses were performed in triplicate.

## 3. Results and Discussion

### 3.1. Bio-Physico-Chemical Properties of Protein Isolates

The aim of this first part of the study was to understand how the different extraction procedures affect the biochemical, physicochemical, and sensory properties of the fish protein isolate. For this, Ca^2+^-ATPase activity, reactive SH content, surface hydrophobicity, TCA-soluble peptide content, lipid oxidation, and fishy odor score were analyzed, and all are reported in Table 1.

#### 3.1.1. Biochemical Properties

The different extraction procedures had considerable influence on the physicochemical features of the fish protein isolate. First, the highest Ca^2+^-ATPase activity was found in NAM prepared from untreated mince. After conventional and modified acid pH-shift processing, the Ca^2+^-ATPase activity of NAM prepared from protein isolates was significantly reduced (*p* < 0.05). This suggested an alteration of the myosin molecule due to its partial unfolding during pH-shifting. Ca^2+^-ATPase activity is a critical indicator, useful in examining the integrity of myofibrillar proteins (especially myosin) in fish muscle [39] and the reduced activity with AAP and APA indicates significant unfolding events occurring in the myofibrillar proteins during extraction. Second, the reactive SH content was the highest in the untreated mince (5.35 mol/10^8^ g protein), which was significantly reduced after pH-shifted treatments (*p* < 0.05) (Table 1). The lowest content of reactive SH was measured in both protein isolates produced using the modified acid pH-shift methods (*p* < 0.05). This finding could imply that SH groups became more prone to oxidation, resulting in higher disulfide bond formation during post-alkaline solubilization. In addition, while sniffing during the post-alkaline pH adjustment, we detected a sulfur-like odor, suggesting the possible formation of H_2_S through strong alkaline process [40], which could be another reason for the reduced available SH content. Third, surface hydrophobicity measurements revealed that the lowest hydrophobicity was found in untreated mince (*p* < 0.05). A low surface hydrophobicity level suggested that the protein was still in its native state and had fewer exposed hydrophobic groups [41]. All acid pH-shifted treatments caused an increase in surface hydrophobicity and the highest level was found in the APA refinement treatment (*p* < 0.05). It was proposed that muscle proteins subjected to acid–alkaline pH adjustment were more denatured, allowing hidden hydrophobic areas to be exposed to the surface. This shows, overall, that the hybrid extraction procedures considerably influenced the structure of the proteins, which may influence functional properties such as gelation.

#### 3.1.2. TCA-Soluble Peptides

The proteolysis of fish proteins induced via enzymatic hydrolysis (autolysis) or acidic hydrolysis can be analyzed by measuring their TCA-soluble peptides [42]. From the results, untreated mackerel mince had the highest content of TCA-soluble peptide (1.59 µmol/g). This indicated that the activity of the endogenous protease residuals, which are abundant in Indian mackerel, could be capable of producing proteolytic breakdown products [36,43]. All pH-shift treatments significantly decreased the TCA-soluble peptide in mackerel mince to varying degrees, and particularly the modified acid pH-shift processes (*p* < 0.05). It can be suggested that these processes had a greater ability to remove the oligopeptides or partially inactivate the proteinase activity during the pH adjustment, resulting in lower TCA-soluble peptides in protein isolates [36,43]. However, the level of proteolysis in the untreated mince and protein isolates can affect their gel-forming abilities, simply because fragmented proteins or peptides are unable to form a robust three-dimensional gel network [8].

#### 3.1.3. Lipid Reduction, Oxidation, and Fishy Odor Analysis

Lipids can influence the gel-forming ability and initiate rancidity if retained with the proteins. As shown in Table 1, the lipid content of untreated mackerel mince was 0.53 g/100 g of sample. After being treated with pH-shift processing, all processes could remove most lipids by more than 80%, indicating the process’s high capacity for separating and removing lipids from raw materials. In particular, the modified acid pH-shift processes, including AAP (82.32% reduction) and APA refinements (84.45% reduction), achieved greater lipid reduction compared to the conventional acid pH-shift process (80.82% reduction). Thus, it is evident that the modified acid treatments can enhance the effectiveness of the removal of lipids. According to Kristinsson et al. [44], lipids become easily separated during the centrifugation process of pH-shift solubilization based on differences in density and solubility.

The TBARS and fishy odor scores of the untreated mince, conventional, and modified acid-made protein isolates are shown in Table 1. These results showed that the highest TBARS value was found in untreated mince, indicating the highest lipid oxidation. Because untreated mince had a higher lipid content, as well as prooxidants like heme proteins, which can both accelerate lipid oxidation, lipid oxidation occurred to a greater extent in untreated mince than in refined protein isolates [45,46]. The occurrence of unpleasant odors in fish flesh, particularly fishy odors, is strongly linked to the development of secondary lipid oxidation products [45]. The large amount of polyunsaturated fatty acids in minced meat makes it prone to oxidation reactions, resulting in high TBARS levels. After pH-shift treatments, all protein isolates had significantly lower TBARS values. Removing lipids and oxidative components from fish mince may lead to lower TBARS levels [46]. When compared to conventional acid-made protein isolate, it was found that both modified acid-made protein isolates had significantly lower TBARS (*p* < 0.05). This was likely correlated with higher lipid reduction (Table 2) and lower levels of total prooxidant heme pigments (Figure 2; see below).

The lower TBARS values were also reflected in the fishy odor analysis. The conventional acid and modified acid pH-shift processes were much more effective at reducing the fishy odor of mackerel mince, resulting in a lower score (*p* < 0.05). Both of the modified acid pH-shift processes produced the protein isolates with the lowest intensity of fishy odor scores when compared to the original acid process (*p* < 0.05). As a consequence, modified acid processing can be used as a deodorizing technique in the production of mackerel protein isolate.

#### 3.1.4. Heme Pigment Removal and Color Characteristics

Heme proteins, particularly myoglobin and hemoglobin, may have a negative effect on the sensory and color characteristics of the mackerel protein isolate due to promoting lipid oxidation and its intrinsic coloring properties [35]. According to the results, the total heme content of untreated mackerel mince was 12.85 μmol/kg (Figure 2a). The quantity of heme pigments in all isolates was significantly reduced after acid pH-shift processing, particularly in both modified acid processes (AAP and APA treatments) (*p* < 0.05), with a heme protein removal of 56% compared to 39% in the conventional acid process. Abdollahi and Undeland [47] reported that heme proteins were eliminated into the first sediment and/or second supernatant during the pH-shift process. To validate this assumption, the absorption myoglobin spectra in the Soret region of discarded supernatant after isoelectric precipitation were measured (Figure 2b). From the result, it was demonstrated that the Soret peak of the discarded supernatant was greater in AAP refinement than in the original process and that the APA process resulted in a much lower peak. The reason for this is that most of the heme proteins have probably already been removed during the first precipitation step (Figure 1). However, the leaching effect of the added water, as well as the destruction of heme proteins under extreme acidic and alkaline pHs, may have resulted in increased myoglobin/hemoglobin removal [32,48].

Based on these results, the color of the protein isolates was measured. For most food applications, a protein isolate without pigments is desirable [49]. The color characteristics of untreated mince and protein isolates recovered via the conventional and modified acid pH-shift processes are shown in Figure 2c. As expected, untreated mince showed the lowest *L** and the highest *a** values, contributing to the lowest whiteness. This was because the mackerel mince had a high level of heme proteins and other pigments. The isolation processes all resulted in an increased *L** value and a reduced *a** value of all recovered proteins due to the removal of pigment, especially for the APA treatment, induced by the two precipitation operations or enhanced heme denaturation. This is in line with Panpipat and Chaijan’s study [32], in which they found that a pH-shift technique caused an increase in *L** value and a decrease in *a** value, resulting in a higher whiteness value of the protein isolates produced from bigeye snapper head by-product. Overall, the modified acid pH-shift process could improve the color properties of mackerel protein isolate more than the conventional process.

### 3.2. Oscillatory Dynamic Rheology

To reveal the rheological properties of untreated mince and protein isolates prepared using conventional and modified acid pH-shift processes, a temperature sweep was carried out to monitor heat-induced gelation behavior, as depicted in Figure 3. The overall G′ curve pattern was similar to that of G″ and always higher, indicating that an elastic gel was formed [50]. As shown in Figure 3a,b, the G′ and G″ curves of all protein isolate samples were different compared to the untreated sample, indicating that the gel formation and the viscoelastic properties of the recovered proteins are influenced by the protein extraction treatments. The G′ and G″ of the untreated sample were constant up to 48 °C, then increased at a high rate until reaching a plateau at around 68 °C, followed by a relatively constant viscoelasticity up to 90 °C. In contrast, the G′ and G″ of all protein isolate samples showed a lag phase up to 60 °C, followed by structure formation up to 90 °C. The rapid increase of G′ after 50 °C probably resulted from an increase in the number of crosslinks in myosin aggregates and the formation of a thermally irreversible gel network [50]. The delayed increases in G′ found for the isolates could be explained by actomyosin aggregation during the extraction procedure prior to heat-induced gelation. The extensive S-S crosslinks indicated by the reduction of SH groups (Table 1) most likely led to an increased aggregated structure that was more resistant to thermal denaturation. Thus, the onset of gelation was delayed. In this study, protein isolates produced via modified acid pH-shifting displayed higher G′ and G″ compared with their original acid counterpart. The result was in line with higher breaking force, deformation, and gel strength (Figure 4a,b; see below). This suggested that the post-alkaline pH-shift of both modified acid processes had a significant impact on the structure of mackerel protein isolates and promoted gelling properties.

Tan δ is used to indicate the ratio of “viscosity” relative to the “elasticity” contributions within a gel network. A tan δ = 0 corresponds to a purely elastic response, and tan δ = 1 is a purely viscous response. If tan δ is within the range of 0 < tan δ < 1, the sample would be considered viscoelastic [42]. From the rheogram, the tan δ of all samples regardless of isolation method was <1.0 (Figure 3c). Thus, all samples performed similarly to an elastic gel [51]. However, the tan δ curves of the protein isolates were different from the untreated samples, indicating the conformation changes of recovered proteins after pH-driven methods. Based on the stronger strength of the intermolecular bonds in the gel structure, the matrix becomes more cohesive as tan δ decreases [52]. At the final temperature of the heat-induced gel, the tan δ of the modified acid-made protein isolate samples was lower than the conventional one’s. This indicated that the protein isolates produced via modified acid pH-shift processes could form a network with a higher elasticity.

### 3.3. Gel Characteristics

#### 3.3.1. Breaking Force, Deformation, Gel Strength, and TPA

The aim of this set of experiments was to further elucidate the gel characteristics of the gels formed, using texture analysis. According to the results presented in Figure 4a, the breaking force and deformation of both gels formed using the protein isolates recovered via the modified acid-aided pH-shift processes, AAP and APA, were significantly higher than those of the conventional acid pH-shift treatment (*p* < 0.05). Additionally, the latter also performed worse in these parameters when compared to the untreated mince (*p* < 0.05). The highest gel strength was observed for gels made of the protein isolate recovered using the APA refinement process (*p* < 0.05). Although decreased Ca^2+^-ATPase activity and a reactive SH group (Table 1) were observed, especially in the APA treatment, the highest gel strength was still found in this protein isolate. Chaijan et al. [48] reported that partial denaturation of the myosin head group can result in the exposing of reactive molecules during the pH-driven protein recovery, which can aid in the creation of a gel structure. According to Somjid et al. [36], protein–protein interactions can be promoted by exposing particular functional groups, such as hydrophilic regions and SH motifs. Moreover, among recovered proteins, the highest hydrophobicity was found in protein isolates recovered using the APA approach (Table 1). This high surface hydrophobicity may have resulted in an increased formation of intra- and inter-hydrophobic crosslinks to form the gel network during the heat treatment. Numerous findings suggested that the large quantity of hydrophobic groups exposed on the surface of the myosin head created via pH-shifting processes might substantially strengthen their hydrophobic interactions [53,54]. In contrast, despite the presence of hydrophobic interactions in the protein isolate recovered using the original acid process, the lowest gel strength was still observed in the gel formed from this starting material (Figure 4b). To understand this phenomenon more research is needed, but it might be speculated that the original acid process may result in greater hydrolysis of the myosin heavy chain (MHC) than the modified acid-produced protein isolate, resulting in a decreased breaking force and deformation of the heat-induced gel. Therefore, the hybrid pH-shift method, particularly APA, may be more effective than the regular acid pH-shift method in improving the gel properties of mackerel protein isolate.

The TPA values of the gels are shown in Table 2. TPA is also one of the main parameters used to reveal the functional characteristics of protein gels. For hardness and springiness, the TPA results were in line with the gel strengths reported in the previous section (Figure 4). Overall, the TPA parameters of both gels made of protein isolates obtained using the two modified acid pH-shift processes were significantly higher than those of the original acid isolation process. The gel prepared with the isolate produced via the APA process appeared to be the optimal method to produce a highly structured protein gel with a high hardness. However, both protein isolates produced using modified acid processes expressed some unique gel textures that differed from each other in their hardness, cohesiveness, and gumminess, and thus may qualify as design-specific gel structures based on the required product parameters, with AAP showing generally less elastic properties.

#### 3.3.2. Expelled Moisture and Color Properties

The expelled moisture is commonly used as an indicator of water-holding capacity, which is an important quality parameter of gelled foods. As depicted in Figure 4d, the refinement approach following APA rendered the gel with the lowest expressible moisture, followed by the AAP refinement and then the original acid treatment (*p* < 0.05). This suggested that the protein network of that gel had the highest water-holding capacity, which was in line with the texture measurements (see previous section) that indicated that the APA extraction approach yielded the most elastic gel network (Figure 4b). The APA extraction may have helped in myosin dissociation, resulting in better dispersion and the development of a compact structure that enabled water molecules to be held tightly in the gel network after thermal gelation. For untreated minced gel, the expelled moisture was higher than for the conventional acid treatment, although its gel strength was higher. This was possibly due to the presence of inactive compounds (e.g., lipids) and aggregates, which can prevent the protein–water interaction. Consequently, the gel matrix could not retain water, leading to high water releases [48].

The whiteness values of all of the gels were higher than those of the non-gelled samples. Cooked gels’ whiteness may be improved by the breakdown of the porphyrin ring of hemoglobin/myoglobin and the aggregation of proteins into three-dimensional networks, which results in light scattering [9]. As shown in Figure 2, the whiteness of all gels showed a similar trend to the whiteness of their protein isolate samples (Table 1). The whiteness of untreated mince gel was lower than that of protein isolate gels, and the whiteness of the protein isolate gel created using both modified acid pH-shift techniques was considerably higher than that of the original acid process (*p* < 0.05). As a result, the modified acid pH-shift processes could improve the whiteness of mackerel protein isolate gel.

#### 3.3.3. FTIR Spectra

Figure 5 shows the FTIR spectra of gels of untreated mince and protein isolates prepared using different acid pH shift processes. The FTIR technique is commonly used to determine the secondary structure of proteins [55]. From the FTIR spectra, all gel samples were similar in their skeleton, indicating that all the pH-shift treatments did not change the number of functional groups. The principal functional groups detected in the FTIR spectra were as follows: amide I, amide II, amide III, amide A, and amide B. The amide I band, a prominent peak at 1640 cm^−1^ in all gel spectra arising from the C=O stretching vibration, indicated a β-sheet preponderance and thus a more stable structure [56]. The amide II band occurred at 1523 cm^−1^, and is assigned to N–H bending and C–N stretching vibrations. Amide III can be noted at 1235 cm^−1^ (N–H deformation and C–N stretching vibration). The N–H stretching motion of amide A was discovered at 3273 cm^−1^, while the –CH stretching vibrations of amide B’s –CH_2_ group were discovered at 2961 cm^−1^. Furthermore, the C–H stretching frequency in FTIR demonstrated the existence of aldehydes. The stretching vibration at 2874 cm^−1^ is produced by combining the distinct C–H frequency with carbonyl oxygen. It indicates that mackerel proteins were not extensively chemically modified by both the conventional and modified acid pH-shift procedures.

#### 3.3.4. Gel Microstructure

The microstructures of unwashed mince gel and protein isolate gels prepared using original acid and modified acid processes were visualized via SEM (Figure 6). From the results, all gel samples had a network structure, suggesting that the gels possessed elastic characteristics. The untreated mince gel displayed a non-uniform gel network with heterogeneous pores and large cavities. In addition, packed spherical proteins arranged in clusters with irregular pores were also found and dispersed throughout the fibrous linkages. This was due to the presence of several compositions in mackerel mince, such as sarcoplasmic proteins, connective tissue, and lipids, which had a detrimental impact on its gel-forming ability and water-holding capacity (Figure 4c) [36]. The gel microstructure of the protein isolates recovered via the three different acid pH-shift treatments appeared different from that of the untreated mince. They exhibited very similar microstructures, particularly the AAP refinement and APA refinement samples. The gels displayed generally quite homogeneous pore distributions as well as cluster structures that were embedded in the network formed by the protein aggregates. However, when compared to its modified acid counterparts, the gel made from protein isolates prepared using the conventional acid pH-shift method had a less porous and more aggregated microstructure, indicating an inferior gel network development, which confirmed the lower values of breaking force and deformation in the resultant gel (Figure 4a).

### 3.4. SDS-PAGE Patterns

SDS-PAGE can be used to demonstrate protein polymerization in non-reducing and reducing conditions to identify potential disulfide bonds in the gel. This can be used to assess the capacity of protein isolates to form gels [6,36]. Furthermore, SDS-PAGE was used to compare the protein patterns of untreated mince and protein isolates generated via conventional and modified acid pH-shift procedures, including non-gelled and gelled samples, in order to evaluate how the protein changed following thermal gelation.

The SDS-PAGE analysis for the protein patterns of untreated mince and protein isolates prepared via conventional and modified acid pH-shift processes, including non-gelled and gelled samples, is presented in Figure 7. In general, myosin heavy chain (MHC) and actin (AC) were discovered to be the two primary bands (approximately 205 and 45 kDa, respectively).

For non-gelled samples, in the absence of βME, there were some visible polymerized proteins at the top of the stacking gel in all samples, particularly after all pH-shift extraction treatments (Figure 7a), indicating the sensitivity of mackerel proteins to aggregation during the recovery process. When comparing the MHC and AC intensity of protein isolates recovered via different acid pH-shift methods, the lowest band intensity was seen in the conventional acid treatment and more research is needed to elucidate why this is the case. However, most polymerized proteins can be split in the presence of βME. This finding showed that the creation of disulfide bonds was not limited to myosin molecules, but also appeared to take place between MHC and AC. Moreover, a new band with a molecular weight of ~150 kDa was clearly seen in all isolated samples using pH-shift processing, with the highest intensity in the conventional acid treatment. The occurrence of low-molecular-weight bands is believed to be a result of some myosin degradation via enzymatic proteolysis [57] as well as protein hydrolysis under acid and alkaline environments [58]. Kristinsson and Liang [59] reported that the occurrence of a new band at 120–160 kDa was probably due to the hydrolysis of MHC.

For the gelled samples, the MHC band as well as certain bands between the molecular weights of 40 and 150 kDa almost completely disappeared after gelation (Figure 7b) in the absence of βME. The polymerized proteins emerged and remained on the stacking gel for all gels, relative to that observed in the untreated mince and protein isolates, presumably due to polymerization during cooking, but their intensities disappeared in the presence of βME. This confirmed that the disulfide bond was one of the links that stabilized the gel networks. Furthermore, other myofibrillar proteins and degraded MHC from pH-shifted samples may participate in the interaction forming the gel network.

## 4. Conclusions

The hybrid pH-shift method can be applied to produce a gel-forming protein isolate from mackerel mince as an alternative refinement process. In comparison with conventional acid pH-shift processing, the modified acid pH-shift method was successful in eliminating lipid and heme proteins, maintaining lipid oxidative stability, and reducing the fishy odor in mackerel mince. As a result, gel properties such as whiteness, gel strength, and water-holding capacity were improved. Overall, the APA refinement approach of the modified acid pH-shift procedure appeared to be the best suited for gel strengthening, as evidenced by its bio-physico-chemical characteristics and gelling properties.

## Figures and Tables

**Figure 1 foods-12-03894-f001:**
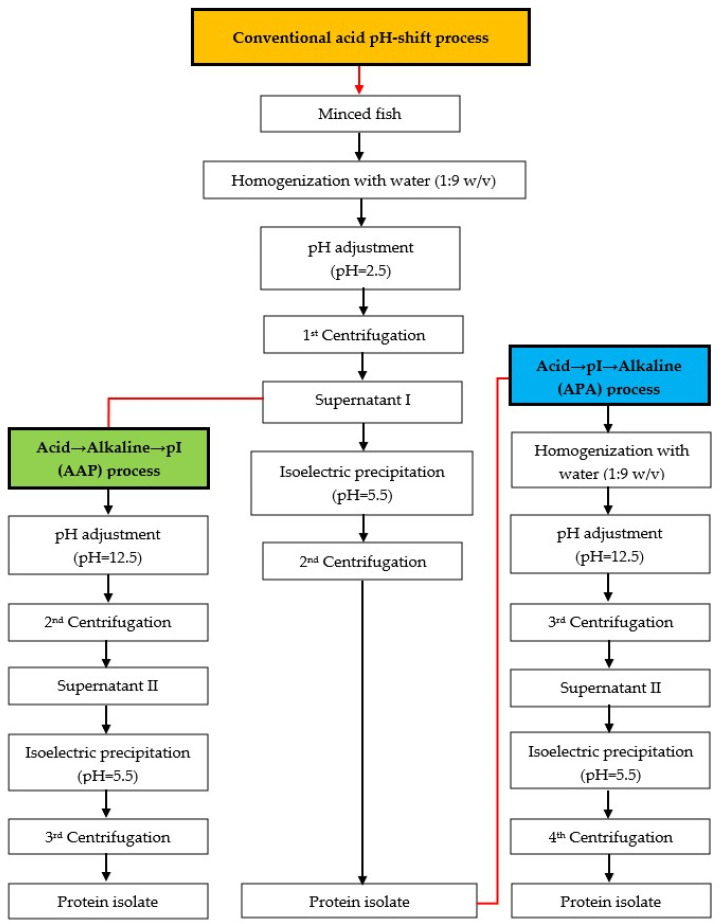
Schematic diagram for producing protein isolates from mackerel mince. Orange-filled box: conventional acid pH-shift process, green-filled box: acid → alkaline → pI (AAP) process, blue-filled box: acid → pI → alkaline (APA) process.

**Figure 2 foods-12-03894-f002:**
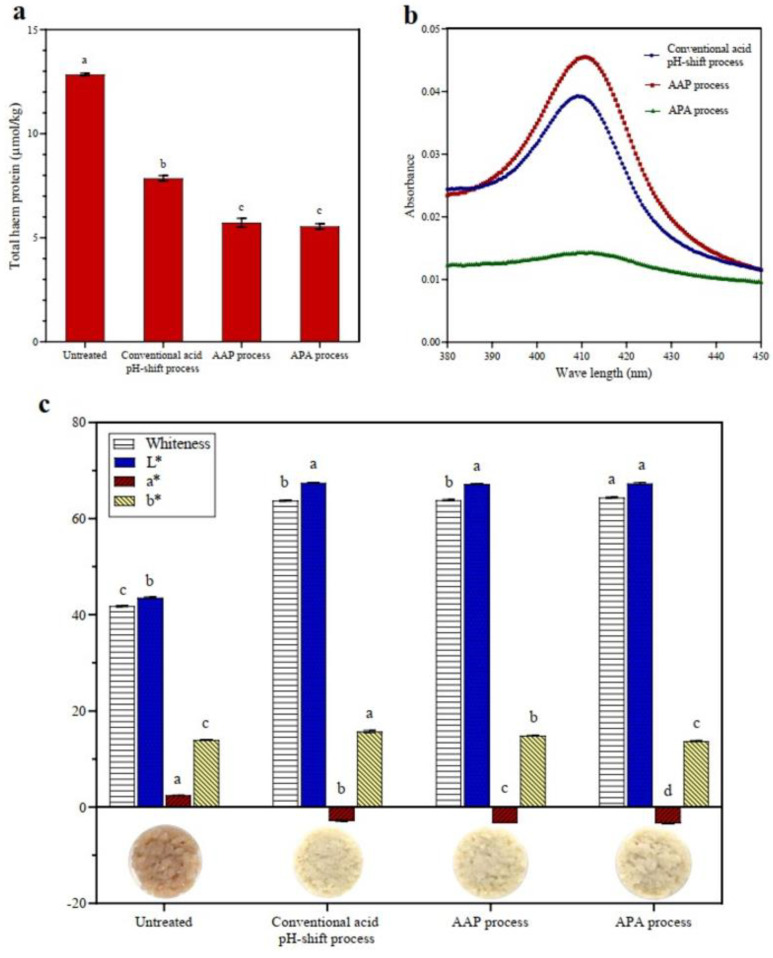
Total heme pigment (**a**), myoglobin spectra in the Soret region of discarded supernatant (**b**), and color characteristics (**c**) of untreated mackerel mince and protein isolates recovered via conventional and modified acid pH-shift processes. Bars are given as means ± standard deviation (*n* = 3). Different letters indicate significant differences between groups (*p* < 0.05).

**Figure 3 foods-12-03894-f003:**
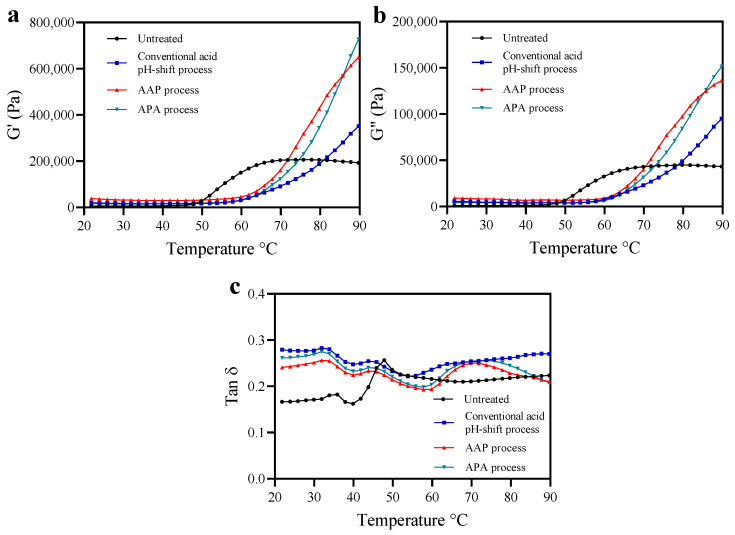
Changes in dynamic viscoelastic behavior (storage modulus (G′; (**a**)), loss modulus (G″; (**b**)) and Tan δ (**c**)) of mackerel untreated mince and protein isolates recovered with conventional and modified acid pH-shift processes.

**Figure 4 foods-12-03894-f004:**
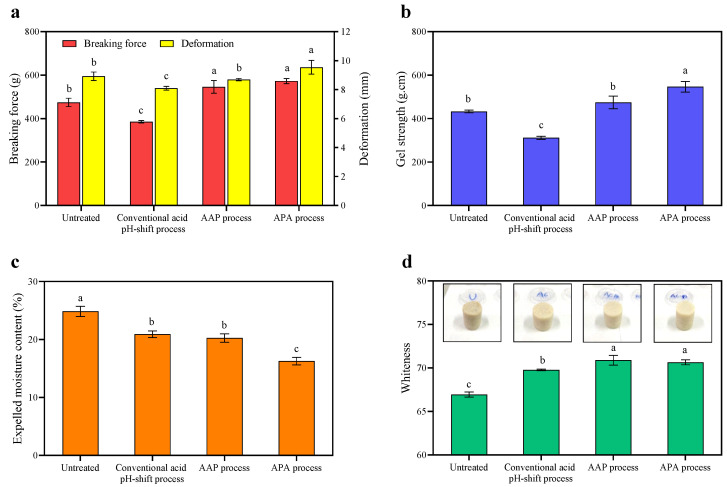
Breaking force and deformation (**a**), gel strength (**b**), expelled moisture content (**c**), and whiteness (**d**) of gels made from untreated mackerel mince and protein isolates recovered via conventional and modified acid pH-shift processes. Bars are given as means ± standard deviation (*n* = 6). Different letters indicate significant differences between groups (*p* < 0.05).

**Figure 5 foods-12-03894-f005:**
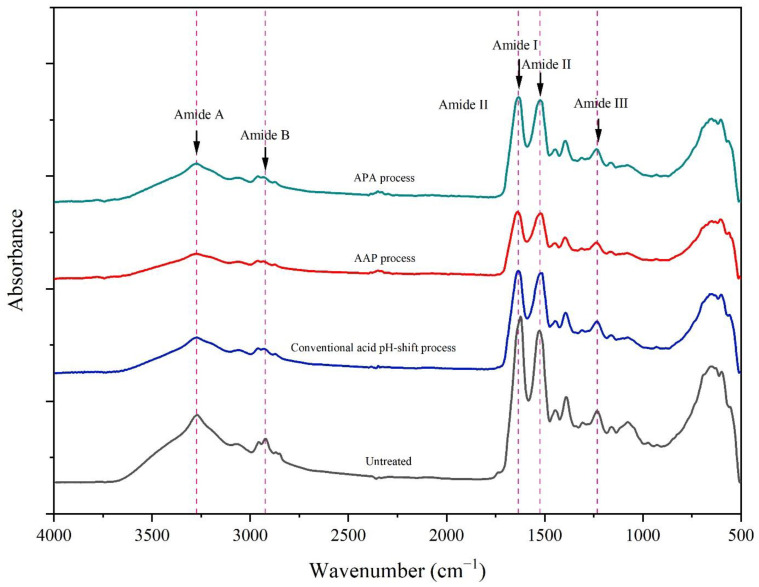
FTIR spectra of gels made from untreated mackerel mince and protein isolates recovered via conventional and modified acid pH-shift processes.

**Figure 6 foods-12-03894-f006:**
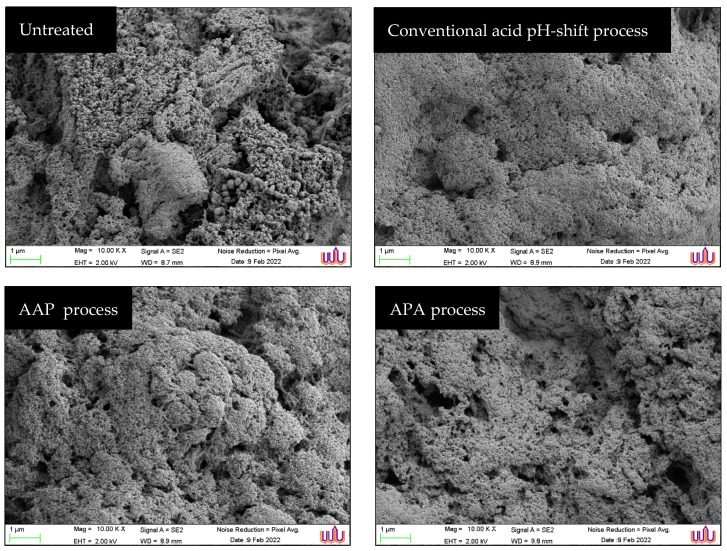
Scanning electron microscopic (SEM) images of gels of untreated mackerel mince and protein isolates recovered via conventional and modified acid pH-shift processes. (Magnification: 10,000×, EHT: 2 kV).

**Figure 7 foods-12-03894-f007:**
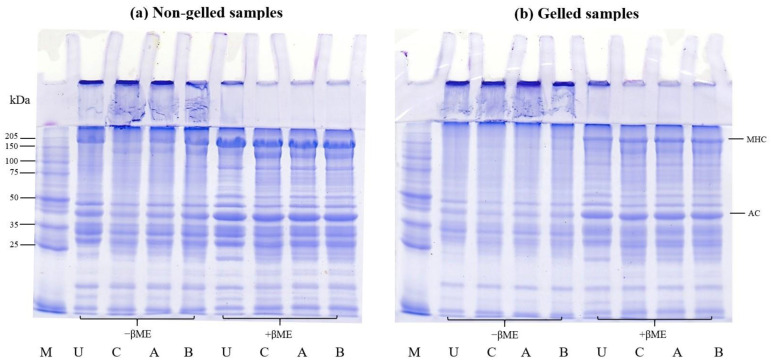
SDS-PAGE patterns before (**a**) and after gelation (**b**) of proteins from untreated mackerel mince (U) and protein isolates prepared via different pH-shift processes: conventional acid-made protein isolate (C), AAP (A), and APA (B). MHC, myosin heavy chain; AC, actin; M, protein markers; −βME, without β-mercaptoethanol; +βME, with β mercaptoethanol.

**Table 1 foods-12-03894-t001:** Ca^2+^-ATPase activity, reactive sulfhydryl (SH) content, surface hydrophobicity, TCA-soluble peptide content, lipid reduction, thiobarbituric acid reactive substances (TBARS), and fishy odor score of untreated mackerel mince and protein isolates recovered using conventional and modified acid pH-shift processes.

Parameter	Untreated	Conventional Acid pH-Shift Process	Modified Acid pH-Shift Process	
			AAP Process	APA Process
Ca^2+^-ATPase activity (µmol/mg protein/min)	2.10 ± 0.02 a	1.48 ± 0.01 b	1.53 ± 0.12 b	1.00 ± 0.04 c
Reactive SH content (mol/10^8^ g protein)	5.35 ± 0.07 a	2.87 ± 0.03 b	2.58 ± 0.08 c	2.57 ± 0.14 c
Surface hydrophobicity (BPB bound (µg))	51.95 ± 4.73 d	105.49 ± 7.07 c	115.10 ± 0.58 bc	121.15 ± 2.54 a
TCA-soluble peptide content (µmol tyrosine/g sample)	1.59 ± 0.27 a	0.53 ± 0.05 b	0.32 ± 0.07 c	0.27 ± 0.06 c
Lipid reduction (%)	-	80.82 ± 1.53 b	82.32 ± 0.49 ab	83.45 ± 1.13 a
TBARS (mg/kg)	9.10 ± 0.38 a	4.59 ± 0.09 b	2.75 ± 0.13 c	2.97 ± 0.10 c
Fishy odor score	7.55 ± 2.56 a	4.51 ± 1.45 b	2.63 ± 1.39 c	2.94 ± 1.65 c

Values are given as mean ± standard deviation (*n* = 3) except for the fishy odor score, which was determined by 10 panelists. Different letters in the same row indicate significant differences (*p* < 0.05). Lipid content of untreated mince was 0.53% of wet weight.

**Table 2 foods-12-03894-t002:** Texture profile analysis (TPA) of gels made from untreated mackerel mince and protein isolates recovered via conventional and modified acid pH-shift processes.

Parameter	Untreated	Conventional Acid pH-Shift Process	Modified Acid pH-Shift Process	
			AAP Process	APA Process
Hardness (N)	19.28 ± 0.62 b	18.36 ± 1.03 c	19.98 ± 0.27 b	27.57 ± 1.15 a
Springiness (cm)	7.44 ± 0.02 b	7.19 ± 0.04 d	7.32 ± 0.05 c	7.54 ± 0.04 a
Cohesiveness	0.44 ± 0.03 b	0.46 ± 0.02 b	0.55 ± 0.00 a	0.48 ± 0.01 b
Gumminess (N)	8.41 ± 0.39 c	8.39 ± 0.06 c	10.96 ± 0.17 b	13.33 ± 0.42 a

Values are given as mean ± standard deviation (*n* = 6). Different letters in the same row indicate significant differences (*p* < 0.05).

## Data Availability

All data are contained within the article.

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
