# Peer review of "The Effect of Multistage Refinement on the Bio-Physico-Chemical Properties and Gel-Forming Ability of Fish Protein Isolates from Mackerel (Rastrelliger kanagurta)"

_foods, 2023, doi:10.3390/foods12213894_

Round 1

Reviewer 1 Report

See the attached document

Some minor editing

Author Response

Manuscript ID: foods-2604940

Title: Effect of Multistage Refinement on Gel Functionality of Fish Protein Isolates from Mackerel (Rastrelliger kanagurta)

The authors indicate in the title that they have calculated the functionality of the gel, but in the manuscript, it is not indicated how this functionality was determined/observed. Mostly are presented the physis and chemical characteristics!

Ans: We used the term "gel functionality" to explain the gelling properties of fish protein isolate. However, to clarify and specify our findings, the title was changed to "Effect of Multistage Refinement on Bio-Physico-Chemical Properties and Gel-Forming Ability of Fish Protein Isolates from Mackerel (Rastrelliger kanagurta)". Also, the objective of the study in the last section in Introduction was also changed to “As a consequence, the current study aims to investigate the effect of a modified acid pH-shift procedure involving post-alkaline processing, simply called the "hybrid pH-shift process", as a refinement strategy to improve the bio-physico-chemical properties and gel-forming ability of a protein isolate from Indian mackerel.

2.1. Fish Sample Preparation: Fish was prepared by hand?

Ans: Yes, fish was prepared by hand. So, it was changed to “The fish were then manually headed, eviscerated, washed, filleted, and skinned respectively.”

Line 109: In brief, mackerel was minced with cold distilled water at a 1:9 (w/v) ratio for 1 min at 20,000 rpm… Was minced or was homogenised?

Ans: It was “homogenised”. So, it was changed to “In brief, mackerel was homogenized with cold distilled water at a 1:9 (w/v) ratio for 1 min at 20,000 rpm using an IKA® homogenizer (Model T25 digital UltraTurrax®, Staufen, Germany).

2.3. Characterization of Protein Isolates and Untreated mince: It is necessary to say untreated mince? Untreated mince should be defined

Ans: The word "untreated mince" was used throughout the text after being defined for the first time as "fresh minced fish" in Section 2.2.

Line 137: Ellman’s reagent (5, 50-dithiobis (2-nitrobenzoic acid); DNTB). Please remove the ) before;

Ans: Ellman's reagent is known as “5, 50-dithiobis (2-nitrobenzoic acid)”, thus we want to maintain it.

2.4.1. Dynamic Rheological Analysis: Line 182:…Which is intended by… paste samples…?

Ans: It was changed to “The rheological behavior of the untreated mince and protein isolates was measured using a rheometer (HAAKE MARS 60; Thermo Fisher Scientific Inc., Yokohama, Japan) with a 35 mm diameter parallel plate [36].

May be the section 2.4.2. Gel Preparation could be before 2.4.1

Ans: The process of dynamic rheological analysis allows one to keep track of the heat-induced protein isolate gelation behavior, which is the fundamental data needed to support the gelation characteristics during thermal gelation in the following phase. Consequently, rheology analysis must be presented prior to thermal gelation.

2.5.2. Water Holding Capacity under Stress

Usually, Water holding capacity (WHC) is the ability of food to hold its own or added water during the application of force, pressure, centrifugation, or heating. Is this according to your meaning? Why use the term expelled moisture and not water holding capacity?

Ans: In this work, we examined the gel's water holding ability by measuring expelled moisture or expressible moisture, as is often done for gel made from fish mince, surimi, and protein isolates. In this study, we used the word "expelled moisture" instead of "expressible moisture" to reduce the similarity index when we checked it using Turnitin.

2.5.3. Fourier Transform Infrared (FTIR) Spectra: The FTIR spectra of freeze-dried gel samples…It is not indicated previously how freeze-dried gel samples were prepared. Could be indicated?

Ans: It was stated that “The gel samples were freeze-dried using a freeze-dryer (FTS systems Inc., Stone Ridge, NY, USA). The FTIR spectra of freeze-dried gel samples were then obtained using a Bruker INVENIO-S FTIR spectrometer (Bruker Co., Ettlingen, Germany) as described by Somjid et al. [6].

3.1. Bio-Physico-Chemical Properties of Protein Isolates

Line 266: …and sensory was analyzed and reported  in Table 1. Suggestion: …and fishy was analysed and reported in Table 1.

Ans: Done.

Table 1. Biochemical properties, TCA-soluble peptide content, lipid reduction, thiobarbituric acid reactive substances (TBARS), and fishy odor score of mackerel untreated mince and protein isolates recovered with conventional and modified acid pH-shift processes. But previously, it is not mentioned biochemical properties! Moreover, it would be better to harmonise the text, for instance the authors use, in some parts of the text, sensory analysis but in others used odour score or even fishy odour score!

Ans: In reality, biochemical properties related to characteristics such as Ca2+-ATPase activity, reactive sulfhydryl (SH) content, surface hydrophobicity, and so on. To avoid confusion, the Table caption was changed to "Table 1. Ca2+-ATPase activity, reactive sulfhydryl (SH) content, surface hydrophobicity, TCA-soluble peptide content, lipid reduction, thiobarbituric acid reactive substances (TBARS), and fishy odor score of untreated mackerel mince and protein isolates recovered with conventional and modified acid pH-shift processes."

Also, the detail in the Section 3.1. Bio-Physico-Chemical Properties of Protein Isolates was modified to “The aim of this first part of the study was to understand how the different extraction procedures affect the biochemical, physicochemical, and sensory properties of the fish protein isolate. For this, Ca2+-ATPase activity, reactive SH content, surface hydrophobicity. TCA-soluble peptide content, lipid oxidation, and fishy odor score was analyzed and reported in Table 1.

In addition, Section 3.1.1 was changed to “3.1.1. Biochemical Properties

For the sensory analysis, the fishy odor score was determined by sensory analysis. Herein, the term "fishy odor" appeared frequently in the text.

Line 249: In addition, sulfur-like odor was noticed…How was this sulphur-like odour noticed? Or observed?

Ans: During the experiment, we detected it by sniffing. So, it was changed to “In addition, while sniffing during the post-alkaline pH adjustment, we detected a sulfur-like odor, suggesting the possible formation of H2S through strong alkaline process [40], which was another reason for the reduced available SH content.

Line 269: It can be suggested that these processes had more potential to remove the oligopeptides or partially inactivate the proteinase activity during the pH adjustment, resulting in lower TCA-soluble peptides in protein isolates. This explanation is based on the literature? If so, a reference is missing!

Ans: The references were added.

Line 271: However, the level of proteolysis in the untreated mince and protein isolates can affect their gel-forming abilities. This sentence could be better explained!

Ans: More information was provided. “However, the level of proteolysis in the untreated mince and protein isolates can affect their gel-forming abilities, simply because fragmented proteins or peptides are unable to form a robust 3-dimensional gel network [8].

3.1.3. Lipid Reduction, Oxidation, and Sensory Analysis: It would be better to precise which is intended by sensory analysis. Usually sensory analysis examines the properties (texture, flavour, taste, appearance, smell, etc.) of a product through the senses (sight, smell, taste, touch and hearing): and when we are only talking about one attribute, it's better to indicate it.

Ans: The "Sensory Analysis" in this section was changed to "Fishy Odor Analysis" to specify the test and harmonize the content as previously recommended. Thank you very much for your thoughtful suggestion.

Line 275: As shown in Table 1, the lipid content of untreated mackerel mince was 0.53 g/100 g of sample. But in such Table the lipid content is not indicated. Is only indicated the Lipid reduction (%)!

Ans: It was originally stated in the Table’s footnote that “Values are given as mean ± standard deviation (n = 3) except for the fishy odor score which was determined by 10 panelists. Different letters in the same row indicate significant differences (p < 0.05). Lipid content of untreated mince was 0.53%, wet weight.

Line 286: The results showed that the highest TBARS value was found in untreated mince, indicating the highest lipid oxidation. It would be better to discuss the effect on protein isolates and the significant difference between conventional acid pH-shift process and the others (AAP process and APA process). Such discussion is missing.

Ans: The issue of the highest untreated mince TBARS was further discussed. “The results showed that the highest TBARS value was found in untreated mince, indicating the highest lipid oxidation. Because untreated mince had a higher lipid content as well as prooxidants like heme proteins, which can both accelerate lipid oxidation, lipid oxidation occurred to a greater extent in untreated mince than in refined protein isolates [45, 46].

In addition, the discussion was initially raised over how processing affected TBARS. “After pH-shift treatments, all protein isolates had significantly lower TBARS values. Removing lipids and oxidative components from fish mince may lead to lower TBARS levels [46]. When compared to conventional acid made protein isolate, it was found that both modified acid-made protein isolates had significantly lower TBARS (p < 0.05). This was likely correlated with higher lipid reduction (Table 2) and lower levels of total prooxidant heme pigments (Figure 2; see below).”

Line 297: …sensory analysis. It is better to indicate about which attribute you are talking about

Ans: It was changed to “The lower TBARS values were also reflected in the fishy odor analysis.”

Line 423: The TPA values of gels are shown in Table 2. This sentence could be before the Table 2.

Ans: It was rearranged per recommendation. 

General comments:

  1. The planning of this work should have been more careful, i.e., identifying the objectives to be achieved and then choosing a set of analytical methodologies to answer the hypothesis formulated.

Ans: Thank you very much. The analytical approaches used in this study were all designed to answer the hypothesis. 

  1. The authors have used an excessive number of determinations and obtained a high number of results. But this makes difficult to discuss and establish correlations between them. Are all these characterisations necessary to the purpose of the present work?

Ans: In order to complete the work, numerous assessments of the bio-physico-chemical characteristics and gel-forming capacity of the protein isolate were made. This was done in order to draw the best conclusions possible from the results, avoiding any implication without proof that can lead to exaggeration. All characterizations aligned with the goal of the current work.

  1. Do these results were necessary and/or indispensable to allow the authors to state that the "hybrid pH-shift process", as a refinement strategy, improved the physic-chemical characteristics and the odour of a protein isolate from Indian mackerel?

Ans: The statement was changed to “As a consequence, the current study aims to investigate the effect of a modified acid pH-shift procedure involving post-alkaline processing, simply called the "hybrid pH-shift process", as a refinement strategy to improve the bio-physico-chemical properties and gel-forming ability of a protein isolate from Indian mackerel.

  1. The calculation of the yield, protein content of the different products as well as the total amount of protein recovered in the different processes were not considered relevant? But such information is also important to take decisions about the thoughts on the "hybrid pH-shift process". A discussion would be important.

Ans: I really appreciate your assessment of this point critically. The yield, such as the dry matter yield and protein yield, will be calculated in the future to account for this.

  1. The authors did not consider the functional properties (e.g., water solubility, water and oil absorption, emulsion capacity and emulsion stability, foaming capacity and foaming stability, bulk density. But such results will be essential for the considerations stated about the "hybrid pH-shift process". Any reason not to have considered them?

Ans: The objective of the study was to examine the gel-forming capabilities of a protein isolate. As a result, the title was altered to reflect the study's purpose. Other protein functional properties can be studied in the future to broaden the potential use and applicability of the refined protein isolate. Thank you very much.

Reviewer 2 Report

Comments for authors

The paper entitled “Effect of Multistage Refinement on the Gel Functionality of 2 Fish Protein Isolate from Mackerel (Rastrelliger kanagurta)” is very interesting research; however, it is necessary to adjust throughout the text:

1.       In result and discussion, section 3.1 there is no discussion of the results with the literature, discuss and complement.

2.       Section 3.1, 3.1.1 and 3.1.2 it is necessary to adequately discuss the results to be able to align with the importance of these, if not adequately discussed, the objective of the was functionality, may be lost.

3.       The images in figure 2c are not visible, it must be improved.

4.       You should select the most relevant results that demonstrate the functionality and its relationship with the treatments since, for example, section 3.4 shows the protein profiles but does not relate it to the functionality or does so in a superficial manner. Likewise, the characterization by electrophoresis are only approximations and therefore cannot be asserted on this basis.

5.       The conclusion is necessary to refer at the objective declared.

6.       The introduction should demonstrate the findings in the literature regarding the characteristics and their relationship to the function of proteins as well as their importance.

7.       More than 40% of the references are outdated because this information shows old research scarce of novelty.

Author Response

Comments for authors

The paper entitled “Effect of Multistage Refinement on the Gel Functionality of 2 Fish Protein Isolate from Mackerel (Rastrelliger kanagurta)” is very interesting research; however, it is necessary to adjust throughout the text:

  1. In result and discussion, section 3.1 there is no discussion of the results with the literature, discuss and complement.

Ans: Section 3.1 was basically an introduction to the topics covered in sections 3.1.1, 3.1.2, and so on.

  1. Section 3.1, 3.1.1 and 3.1.2 it is necessary to adequately discuss the results to be able to align with the importance of these, if not adequately discussed, the objective of the was functionality, may be lost.

Ans: The bio-physico-chemical characteristics and fishy odor of the protein isolates were extensively discussed in this section. The gel-forming ability, which referred to gel functionality, was then discussed in the next section.

  1. The images in figure 2c are not visible, it must be improved.

Ans: The images in Fig. 2c can be visible (see attachment). It could have been due to a technical problem in the file for the reviewer.

  1. You should select the most relevant results that demonstrate the functionality and its relationship with the treatments since, for example, section 3.4 shows the protein profiles but does not relate it to the functionality or does so in a superficial manner. Likewise, the characterization by electrophoresis are only approximations and therefore cannot be asserted on this basis.

Ans: Section 3.4, SDS-PAGE can be used to demonstrate protein polymerization in non-reducing and reducing conditions to identify potential disulfide bonds in the gel. This can be used to assess the capacity of protein isolates to form gels. Furthermore, SDS-PAGE was used to compare the protein patterns of untreated mince and protein isolates generated by conventional and modified acid pH-shift procedures, including non-gelled and gelled samples, in order to evaluate how the protein changed following thermal gelation.

  1. The conclusion is necessary to refer at the objective declared.

Ans: It is absolutely correct to draw the conclusion after considering the purpose. The revised objective of this study was stated in the introduction. “As a consequence, the current study aims to investigate the effect of  a modified acid pH-shift procedure involving post-alkaline processing, simply called the "hybrid pH-shift process", as a refinement strategy to improve the bio-physico-chemical properties and gel-forming ability of a protein isolate from Indian mackerel.” As a consequence, the conclusion was established to be consistent with the purpose. “The hybrid pH-shift method can be applied to produce gel-forming protein isolate from mackerel mince as an alternate refinement process. In comparison with conventional acid pH-shift processing, the modified acid pH-shift method was successful in eliminating lipid and heme proteins, maintaining lipid oxidative stability, and reducing fishy odor   in mackerel mince. As a result, gel properties such as whiteness, gel strength, and water holding capacity were improved. Overall, the APA refinement approach of the modified acid pH-shift procedure appeared to be the best suited for gel strengthening, as evidenced by bio-physico-chemical characteristics and gelling properties.

  1. The introduction should demonstrate the findings in the literature regarding the characteristics and their relationship to the function of proteins as well as their importance.

Ans: The introduction was cautious, beginning with the importance of fish resources for human nutrition and progressing to the challenges associated with the production of protein isolates or products from dark-fleshed fish such as mackerel. The prior method of solving difficulties utilizing the pH-shift procedure was then informed, followed by the benefits and drawbacks or limitations of each version of the pH-shift procedure, leading to the formulation of the research subject matter. The proposed strategy offered in this study was mentioned in the objective in the last section to solve the remaining problem. 

  1. More than 40% of the references are outdated because this information shows old research scarce of novelty.

Ans: All of the references cited in this work were relevant to the research background, methods, and discussion. Of course, some of them are dated but they were the methods paper that needed to be acknowledged in the original work. The reference list has been updated. This manuscript contained 37 papers published between 2015 and 2023.The work’s novelty was evaluated not only by the updated references, but also by the introduction of a new idea or a unique perspective that adds to the current knowledge in a certain field of study. It entails bringing something new and unique to the table that hasn't been done before, or addressing an existing issue in a novel and innovative way. Thank you very much.

Round 2

Reviewer 1 Report

The answers were satisfactory.

Author Response

Thank you very much.

Reviewer 2 Report

My decision is to decline the second review, although in the opinion I decide to reject since the authors decided to ignore my comments and above all only present answers and in that sense my review does not support the improvement of the document.

Author Response

Dear Academic Editor,

I sincerely appreciate it. Since Reviewer 2 simply made general comments, he/she did not make any particular criticisms. So we did our best to provide thoughtful responses to each query. But since Question 4 seems to be the most explicit, we added some details in the revised version. We've all already provided answers to the other questions in the prior revision. We were certain that every response satisfied the reviewer's requirements. We sincerely hope you get our point.

The responses for all questions are listed below.

Comments for authors

The paper entitled “Effect of Multistage Refinement on the Gel Functionality of 2 Fish Protein Isolate from Mackerel (Rastrelliger kanagurta)” is very interesting research; however, it is necessary to adjust throughout the text:

  1. In result and discussion, section 3.1 there is no discussion of the results with the literature, discuss and complement.

Ans: Section 3.1 was basically an introduction to the topics covered in sections 3.1.1, 3.1.2, and so on.

Ans to Academic Editor: We explain the lack of discussion in section 3.1 by noting that it served only as an introduction to parts 3.1.1, 3.1.2, and so on.

  1. Section 3.1, 3.1.1 and 3.1.2 it is necessary to adequately discuss the results to be able to align with the importance of these, if not adequately discussed, the objective of the was functionality, may be lost.

Ans: The bio-physico-chemical characteristics and fishy odor of the protein isolates were extensively discussed in this section. The gel-forming ability, which referred to gel functionality, was then discussed in the next section.

Ans to Academic Editor:  We tried our best in the discussion. In the revised version, we stated that the bio-physico-chemical characteristics and fishy odor of the protein isolates were extensively discussed in this section. The gel-forming ability, which referred to gel functionality, was then discussed in the next section.

  1. The images in figure 2c are not visible, it must be improved.

Ans: The images in Fig. 2c can be visible (see below). It could have been due to a technical problem in the file for the reviewer.

Ans to Academic Editor:  The image in the word file is fine.

  1. You should select the most relevant results that demonstrate the functionality and its relationship with the treatments since, for example, section 3.4 shows the protein profiles but does not relate it to the functionality or does so in a superficial manner. Likewise, the characterization by electrophoresis are only approximations and therefore cannot be asserted on this basis.

Ans: Section 3.4, SDS-PAGE can be used to demonstrate protein polymerization in non-reducing and reducing conditions to identify potential disulfide bonds in the gel. This can be used to assess the capacity of protein isolates to form gels. Furthermore, SDS-PAGE was used to compare the protein patterns of untreated mince and protein isolates generated by conventional and modified acid pH-shift procedures, including non-gelled and gelled samples, in order to evaluate how the protein changed following thermal gelation.

Ans to Academic Editor:  The statement was provided as a brief introduction in Section 3.4 to connect the SDS-PAGE protein pattern to gel formation.

  1. The conclusion is necessary to refer at the objective declared.

Ans: It is absolutely correct to draw the conclusion after considering the purpose. The revised objective of this study was stated in the introduction. “As a consequence, the current study aims to investigate the effect of  a modified acid pH-shift procedure involving post-alkaline processing, simply called the "hybrid pH-shift process", as a refinement strategy to improve the bio-physico-chemical properties and gel-forming ability of a protein isolate from Indian mackerel.” As a consequence, the conclusion was established to be consistent with the purpose. “The hybrid pH-shift method can be applied to produce gel-forming protein isolate from mackerel mince as an alternate refinement process. In comparison with conventional acid pH-shift processing, the modified acid pH-shift method was successful in eliminating lipid and heme proteins, maintaining lipid oxidative stability, and reducing fishy odor   in mackerel mince. As a result, gel properties such as whiteness, gel strength, and water holding capacity were improved. Overall, the APA refinement approach of the modified acid pH-shift procedure appeared to be the best suited for gel strengthening, as evidenced by bio-physico-chemical characteristics and gelling properties.

Ans to Academic Editor:  We came to a conclusion that addressed the study question and was in line with the goal.

  1. The introduction should demonstrate the findings in the literature regarding the characteristics and their relationship to the function of proteins as well as their importance.

Ans: The introduction was cautious, beginning with the importance of fish resources for human nutrition and progressing to the challenges associated with the production of protein isolates or products from dark-fleshed fish such as mackerel. The prior method of solving difficulties utilizing the pH-shift procedure was then informed, followed by the benefits and drawbacks or limitations of each version of the pH-shift procedure, leading to the formulation of the research subject matter. The proposed strategy offered in this study was mentioned in the objective in the last section to solve the remaining problem. 

Ans to Academic Editor:  We addressed the introduction thoroughly, drawing on relevant literature. All the information provided in the introduction can be used to set up research questions and objectives.

  1. More than 40% of the references are outdated because this information shows old research scarce of novelty.

Ans: All of the references cited in this work were relevant to the research background, methods, and discussion. Of course, some of them are dated but they were the methods paper that needed to be acknowledged in the original work. The reference list has been updated. This manuscript contained 37 papers published between 2015 and 2023.The work’s novelty was evaluated not only by the updated references, but also by the introduction of a new idea or a unique perspective that adds to the current knowledge in a certain field of study. It entails bringing something new and unique to the table that hasn't been done before, or addressing an existing issue in a novel and innovative way. Thank you very much.

Ans to Academic Editor:  The reference has been updated. The reason why some old papers are required was mentioned, which is because they are method papers.
